# The Effects of Multi-Stage Homogenizations on the Microstructures and Mechanical Properties of Al–Zn–Mg–Zr–Sc Alloys

Yang-Chun Chiu [1,2] 🄳, Tse-An Pan [1], Mien-Chung Chen [1], Jun-Wei Zhang [1], Hui-Yun Bor [1] and Sheng-Long Lee [1,*]

1   Institute of Materials Science and Engineering, National Central University, Jhongli 32001, Taiwan; 106389001@cc.ncu.edu.tw (Y.-C.C.); 109389001@cc.ncu.edu.tw (T.-A.P.); 108389002@cc.ncu.edu.tw (M.-C.C.); wangsevenegg@gmail.com (J.-W.Z.); hohh@csnet.gov.tw (H.-Y.B.)
2   Department of Mechanical Engineering, Minghsin University of Science and Technology, Hsinchu 30401, Taiwan
*   Correspondence: shenglon@cc.ncu.edu.tw

**Abstract:** This study is aimed at exploring the effects of multi-stage homogenization and trace amounts of Zr and Sc on the microstructures, mechanical properties, and recrystallization of Al–4.5Zn–1.5Mg alloys. The mechanical properties of the AA7005 aluminum alloys after the T6 heat treatment were evaluated through a hardness test and tensile test. The microstructures were analyzed by an optical microscope (OM), a differential scanning calorimeter (DSC), a transmission electron microscope (TEM), a scanning electron microscope (SEM), and electron backscattered diffraction (EBSD). The results show that the grain refinement effect of the as-cast, homogenized, and recrystallized Al–4.5Zn–1.5Mg alloy containing 0.05Sc (wt%) after the T6 heat treatment was more significant than that of the alloy containing 0.1Zr (wt%). In addition, compared with the aforementioned one-stage homogenization heat treatment, the two-stage homogenization made the dispersed grain phase (Al₃Zr/Al₃Sc) smaller. As a result, the T6 mechanical strength of the alloy after the two-stage homogenization heat treatment was better than that of the contrastive alloy after the one-stage homogenization heat treatment. However, the two different homogenization heat treatments caused a greater divergence between the sizes of the dispersed grain phases of the Al–4.5Zn–1.5Mg alloys containing Zr than between the sizes of the dispersed grain phases of the alloys containing Sc. Therefore, after the two-stage homogenization heat treatment, the alloy with 0.1Zr (wt%) promoted the mechanical properties better than the alloy with 0.05Sc (wt%).

**Keywords:** Al–Zn–Mg alloy; zirconium; scandium; multi-stage homogenizations; recrystallized microstructures

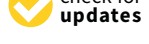

## 1. Introduction

The 7000 series Al–Zn–Mg (–Cu) alloy is a forged, heat-treated and high-strength aluminum alloy, which mainly achieves the precipitation strengthening effect by precipitating the second phase of the integration or semi-integration in the aluminum matrix. It is the strongest alloy among all aluminum alloys series and has high specific strength, high fracture toughness and fatigue resistance; it has been widely used in the aerospace and military industries [1,2]. In the Cu-free 7000 series Al–Zn–Mg alloy, its strength is lower than that of the high-strength Al–Zn–Mg–Cu (e.g., 7075) aluminum alloy, but it does not produce a low melting point eutectic phase S(Al₂CuMg). Therefore, its extrusion properties are better than those of the Cu alloy, and its welding and corrosion resistance are also among the best in the 7000 series alloys. It is one of the main alloys for making large thin walls with high-precision complex solid or hollow structural sections. It has been commonly used as the main material for large lightweight transportation vehicles, such as the manufacture of the extruded sections and plates for high-speed train bodies [3].

It is worthwhile to research and develop a new type of alloy with good weldability and strength without adding Cu atoms. When the aluminum alloy contains trace transition elements such as Mn, Cr, Zr, Sc, etc., whose partition ratios are greater than 1, a peritectic reaction is prone to occur during the casting process and to segregate the grains [4,5]. During the homogenization heat treatment, dispersed phase grains such as $Al_6Mn$, $Al_4Mn$, E-$(Cr_2Mg_3Al_{18})$, $Al_7Cr$, $Al_3Zr$, $Al_3Sc$, etc., will precipitate in the grains. These dispersed phase grains have high temperature thermal stability [4], and can effectively inhibit the dislocations and grain boundary migrations of the alloy. In addition to increasing the recrystallization temperature of the alloy [5,6], they can also significantly inhibit the growth of crystal grains [7]. Cr is prone to form E-$(Cr_2Mg_3Al_{18})$, which will increase the quenching sensitivity of the alloy and affect the properties of the alloy [8]. Among the trace elements, the inhibitory effects of $Al_3Zr$ and $Al_3Sc$ are the best [4], so these two elements (Zr, Sc) were selected for the study. Zirconium has many benefits in aluminum alloys. The aluminum alloy matrix containing zirconium can easily form fine $Al_3Zr$ dispersed grains at high stable temperature, which can inhibit dislocations or grain boundary migrations and increase the recrystallization temperature of the alloy [9]. The addition of zirconium to the aluminum alloys also has a significant grain refining effect [7]. Zr has higher pore bonding energy than Mg and Zn. It can occupy the pores to suppress the main strengthening $\eta'$ phase ($MgZn_2$) from prematurely precipitating in the solution quenching process, and transform the $\eta'$ phase into the $\eta$ phase to reduce the quenching sensitivity of the alloy [9,10]. Moreover, with the addition of scandium, it is extremely easy to form very small $Al_3Sc$ grains in the aluminum alloy matrix, which also have similar effects. When the adding equal amounts of Zr and Sc, certain effects of the $Al_3Sc$ grains are enhanced [7,9,11,12].

The process of the alloy homogenization heat treatment is performed at a high temperature, which is below the eutectic temperature, and its purpose is mainly to reduce the micro-segregation of the as-cast alloy. Traditionally, in order to improve the solubility and accelerate the diffusion rate of the segregated elements, the homogenization will be performed at a temperature as high as possible. Due to the high temperature, it is easy to cause the aforementioned high-temperature thermally stable phase to coarsen, and to lower the precipitation volume fraction. This will affect the subsequent thermal processing and the function of suppressing the dislocations or grain boundary migrations in the recrystallization process [13]. Due to the high-temperature thermally stable phase formed by the transition element during the homogenization, its morphology, size, and distribution have a significant effect on the properties of the alloys [14]. Two or more stages of homogenization heat treatments are often used [15–17]. They are combined with low temperature and high temperature homogenization processes to produce a dense and fine high-temperature thermally stable phase distribution to optimize the microstructures and mechanical properties of the aluminum alloys [13].

This study intends to combine trace zirconium, scandium, and homogenization heat treatment processes to explore the influences on the microstructures and mechanical properties of the Al–4.5Zn–1.5Mg (AA7005) alloy through the T6 heat treatment, with the intention of improving the mechanical strength and the T6 microstructures. As far as the collected literature is concerned [4,13,15,16,18,19], no related research on this subject has been published.

## 2. Materials and Methods

The pure aluminum ingot (99.8%) was placed in a 750 °C resistance furnace crucible. After melting, pure Zn (99.9%), pure Mg (99.9%), aluminum zirconium master alloy (Al–10Zr), etc., were added according to the alloy design. After the master alloy (Al–1.12Sc) had been fully melted and homogenized, it was degassed with pure argon for 30 min, left to stand for 10 min, deslagged, and cast into a mold (125 mm × 100 mm × 25 mm) preheated to 350 °C in advance.

The compositions of the experimental alloy samples were analyzed by the optical emission spectrometer (OES). As shown in Table 1, samples were designated as alloy

A(0.1Zr): (Al–4.47Zn–1.45Mg–0.13Zr) and alloy B(0.05Sc): (Al–4.46Zn–1.43Mg–0.05Sc). The main composition was in compliance with the AA7005 component specifications [20]. The cast alloys were subjected to two different homogenization heat treatments: the one-stage homogenization (470 °C) for 24 h, and the two-stage homogenization (410 °C) for 3 h and then (470 °C) for 20 h. The homogenized alloys were hot-rolled at 420 °C (25 mm to 3 mm, deformation 88%), annealing heat-treated at 420 °C for 2.5 h, and then cold-rolled (3 mm to 2.1 mm, cold worked 30%). Finally, the alloys were subjected to the T6 heat treatment. The sequence was as follows: the solution treatment (470 °C) for 1 h; water quenching [21] and then artificial aging (120 °C) for 24 h [21].

**Table 1.** The composition of the experimental alloy (wt%).

| Composition Alloy | Zn | Mg | Fe | Si | Zr | Cu | Sc | Al |
|---|---|---|---|---|---|---|---|---|
| Alloy A(0.1Zr) | 4.47 | 1.45 | <0.01 | <0.01 | 0.13 | <0.1 | N.D. | Rem. |
| Alloy B(0.05Sc) | 4.46 | 1.43 | <0.01 | <0.01 | <0.01 | <0.1 | 0.05 | Rem. |

N.D.: Non-detectable; Rem.: Remainder.

An optical microscope (OM, Olympus BX60M), a field emission scanning transmission electron microscope (TEM, ULTRA-HRTEM, JEOL JEM-ARM200FTH) and a field emission gun scanning electron microscope (Thermal Field Emission Scanning Electron Microscope (FEG-SEM) + electron backscattered diffraction (EBSD), JEOL JSM-7800F Prime) were utilized to observe the microstructures. Image analysis software (ImageJ) was used to analyze the grain size by the intercept method, and the dispersed grain size of the TEM images. The composition of the dispersed grains was analyzed with an Energy-dispersive X-ray spectroscopy (EDS). An EBSD-assisted analysis software (AZtec) was applied to analyze the diameters of the grains, the average grain area, and grain boundary angle distribution. EBSD-assisted analysis software (Channel5 Tango) was applied to quantify the microstructures of the alloy recrystallization (grain boundary angle >15°).

A high temperature differential scanning calorimeter (HTDSC, NETZSCH, STA 449 F3) was used to analyze the phase change of the T6 test strip, weighing about 20 mg at 50 °C~630 °C (The heating rate was 10 °C/min), and an Origin Pro was utilized to calculate the DSC curves.

The hardness of the alloys was measured with a Vickers hardness machine (HV). The tensile test of the T6 alloy was measured with a 10-ton MTS(metre–tonne–second) closed loop hydraulic servo control universal test machine at room temperature and at a constant strain rate ($1.3 \times 10^{-3}$ s$^{-1}$). The constant extension rate of the tensile testing was 0.2 mm/min from the beginning. When the strain was stretched to 1%, the rate was increased to the final rate of 2 mm/min until the test piece broke. The test piece specifications are shown in Figure 1 [22].

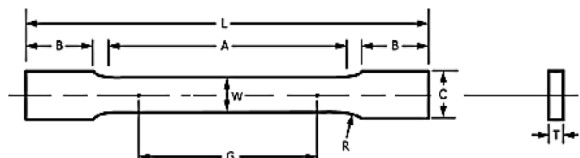

G：25mm±0.1mm,　W：6mm±0.1mm,　L：100mm,　C：10mm, T：2.25mm ,　R：6mm,　A：32mm,　B：30mm

**Figure 1.** ASTM B557M standard tensile bar.

## 3. Results and Discussion

This study aimed to explore how multi-stage homogenization inhibits the recrystallization of the commercial AA7005 (Al–4.5ZZn–1.5Mg) aluminum alloy and upgrades its mechanical properties. Certainly, the designed and experimental alloys cannot fully or satisfactorily replace the functions of the AA7005 alloy. For one thing, the tensile test cannot cover the complete mechanical properties. For another, the chemical and physical

properties of the alloys will need to be studied further. However, this study might offer a valuable reference for future researchers.

### 3.1. Microstructural Observation

In the as-cast state of alloy A(0.1Zr) and alloy B(0.05Sc), the dendrite structures inside the grains can be clearly observed in Figure 2a,b, respectively. Most of the dendritic structures were eliminated through the one-stage homogenization heat treatment (470 °C * 24 h) as shown in Figure 2c,d. Similarly, the dendrite structures in the as-cast state were fully eliminated through the two-stage homogenization heat treatment (410 °C * 3 h + 470 °C * 20 h). The grain diameter was measured by the intercept method, as shown in Figure 2e,f. The average grain diameters of alloy A(0.1Zr) and alloy B(0.05Sc) were approximately 54.0 μm (standard deviation was 10.3 μm) and 49.0 μm (standard deviation was 8.5 μm), respectively.

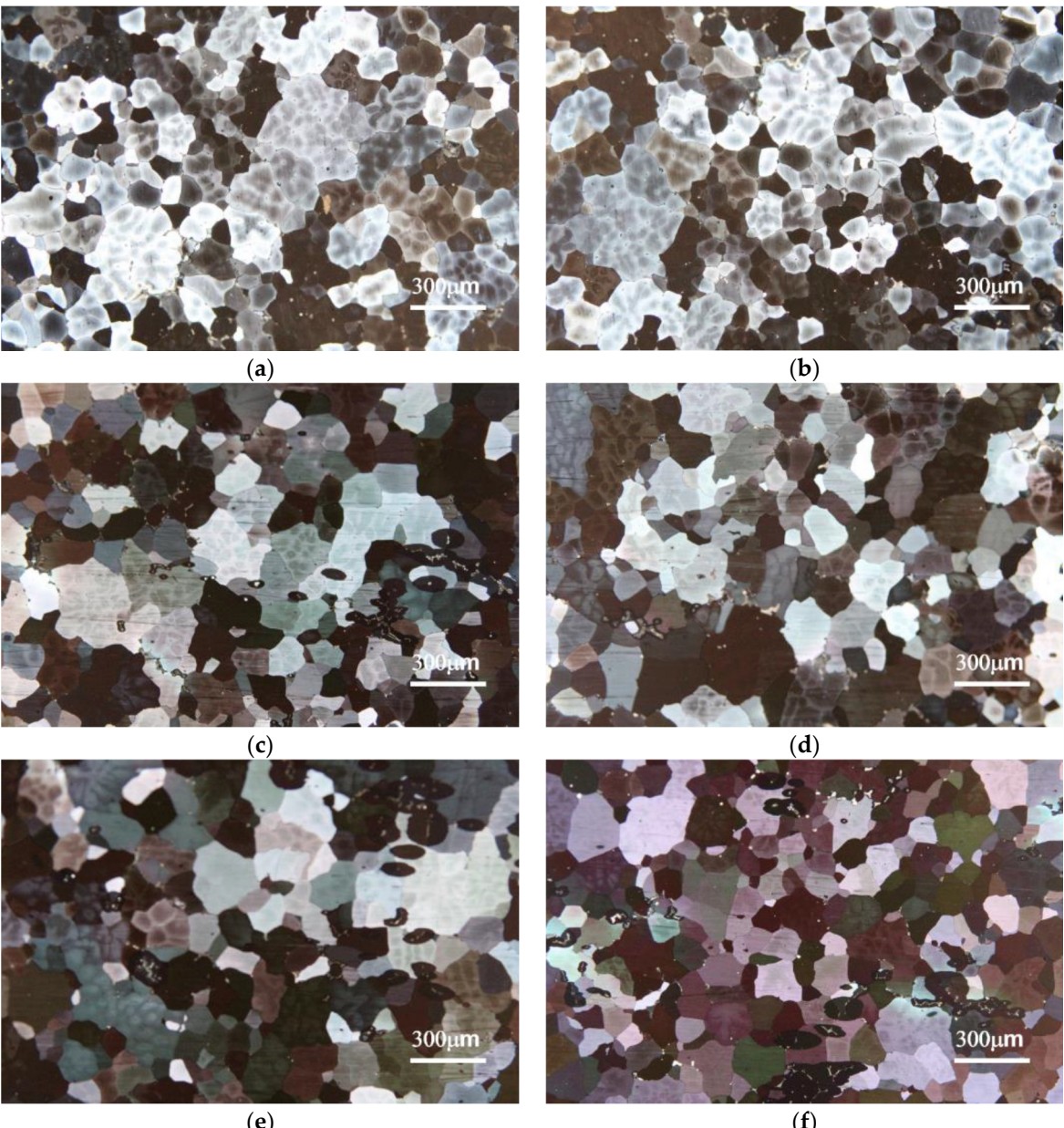

**Figure 2.** The optical microscope images of (**a**) as-cast alloy A(0.1Zr); (**b**) as-cast alloy B(0.05Sc); (**c**) 1-Hom. alloy A(0.1Zr); (**d**) 1-Hom. alloy B(0.05Sc); (**e**) 2-Hom. alloy A(0.1Zr); (**f**) 2-Hom. alloy B(0.05Sc).

By the measurement of the intercept method, after one-stage and two-stage homogenization heat treatments, the grain diameters of the A(0.1Zr) alloys were about 58.0 μm (standard deviation was 12.2 μm) and 55.0 μm (standard deviation was 9.5 μm), respectively; the grain diameters of the B(0.05Sc) alloys were about 54.0 μm (standard deviation was 7.5 μm) and 51.0 μm (standard deviation was 14.3 μm), respectively. Therefore, it can be seen that the grain refining of alloy B(0.05Sc) was better than that of alloy A(0.1Zr), and that the grains of both alloy A(0.1Zr) and alloy B(0.05Sc) were smaller after the two-stage homogenization heat treatment than those after the one-stage homogenization heat treatment.

The aluminum alloys containing trace transition elements such as Zr and Sc were prone to precipitate thermally stable dispersed phases ($Al_3Zr$, $Al_3Sc$) [5]. The TEM microstructures of alloy A(0.1Zr) and alloy B(0.05Sc) after the one-stage homogenization heat treatment are shown in Figure 3a,b, respectively. The dispersed precipitation phases of the spherical grains of alloy A(0.1Zr) and alloy B(0.05Sc) in the aluminum matrix can be seen. Through the TEM–EDS analysis, the components were $Al_3Zr$ and $Al_3Sc$, respectively, as shown in Figure 3e,f; Figure 3c,d are the TEM microstructures of alloy A(0.1Zr) and alloy B(0.05Sc), respectively, after the two-stage homogenization heat treatment.

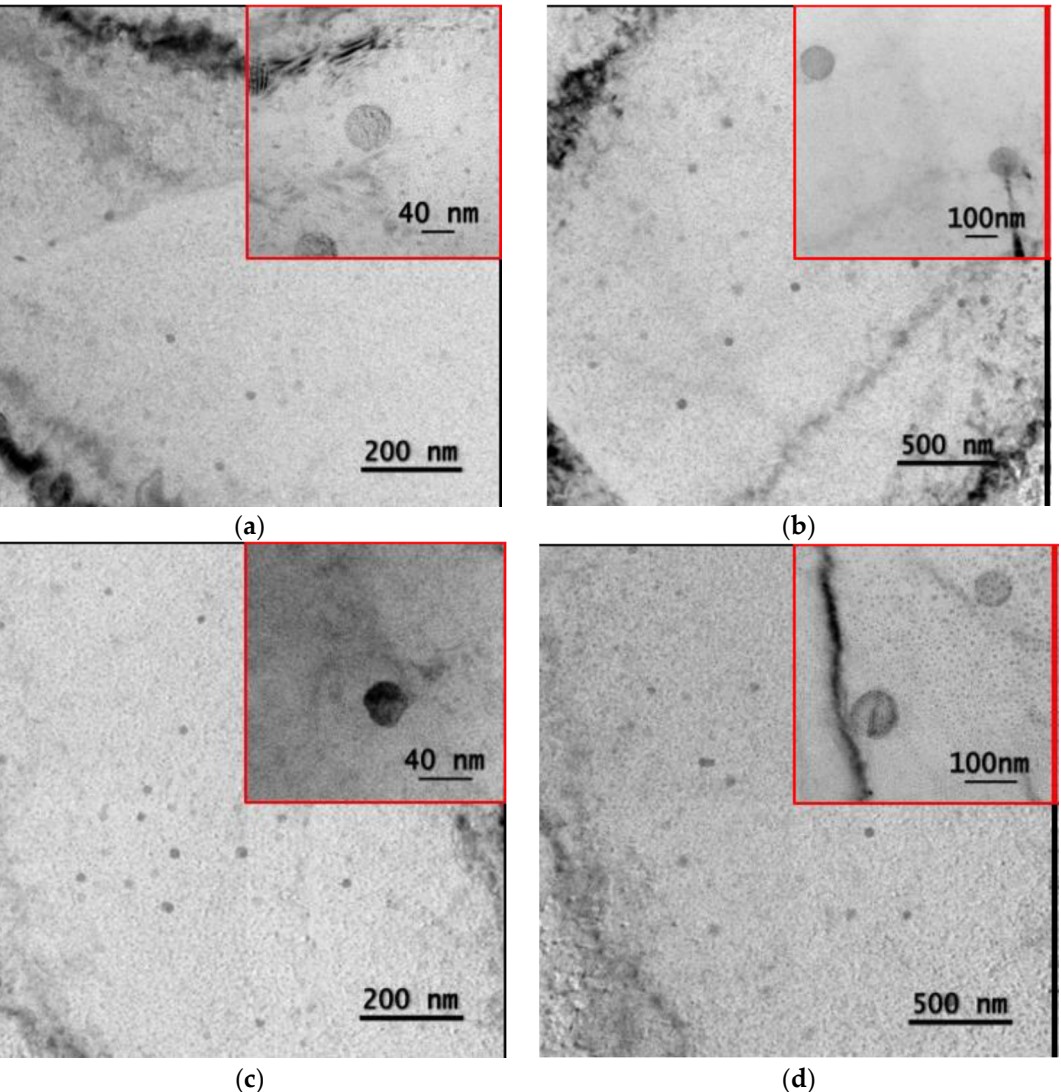

**Figure 3.** *Cont.*

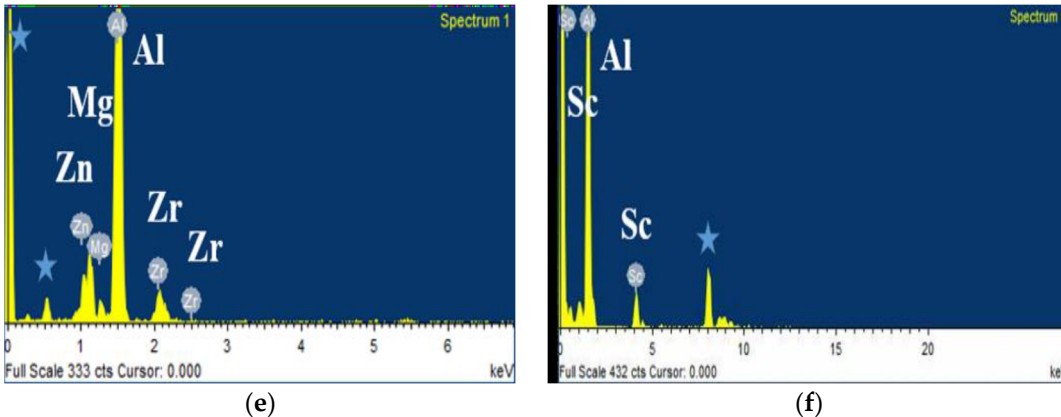

**Figure 3.** Microstructures observation showing $Al_3Zr$ (Spherical-like) and $Al_3Sc$ (Spherical-like) by transmission electron microscope (TEM) analysis of (**a**) 1-Hom. alloy A(0.1Zr); (**b**) 1-Hom. alloy B(0.05Sc); (**c**) 2-Hom. alloy A(0.1Zr); (**d**) 2-Hom. alloy B(0.05Sc); (**e**) EDS analysis of spherical-like particle in alloy A(0.1Zr); (**f**) EDS analysis of spherical-like particle in alloy B(0.05Sc) (asterisks indicate that peaks are possibly omitted).

Comparatively, as shown in Figure 3a,c, after different homogenizations, the grain diameters of the $Al_3Zr$ dispersed phases of Zr-containing alloy A(0.1Zr) can be observed through the image software (ImageJ) analysis. The $Al_3Zr$ after the two-stage homogenization was smaller than that after the one-stage homogenization, and the diameters were about 54 nm and 34 nm, respectively, with a difference of about 40%. In a lower magnification observation, the $Al_3Zr$ distribution after the two-stage homogenization was denser than that after the one-stage homogenization treatment; however, although the grain diameters of the $Al_3Zr$ dispersed phases of the Sc-containing alloy B(0.05Sc) after the two-stage homogenization treatment was smaller than that after the one-stage homogenization (Figure 3d), their diameters were about 81 nm and 72 nm respectively, only with a difference of about 10%. To sum up, the different values of the B(0.05Sc) alloys are not as obvious as those of the A(0.1Zr) alloys, and there were no great differences between the levels (81 nm and 72 nm) of the distribution density of the grains precipitating in the B(0.05Sc) alloys.

The phenomena probably occurred because the precipitation temperatures of $Al_3Sc$ and $Al_3Zr$ were between 250 °C~350 °C [21] and 350 °C~410 °C [9], respectively. When the alloys were subjected to the two-stage homogenization heat treatment (410 °C for 3 h and 470 °C for 20 h), 410 °C was a relatively low temperature for the precipitation of $Al_3Zr$. Therefore, when alloy A(0.1Zr) went through the two-stage homogenization heat treatment, the $Al_3Zr$ grains obtained were significantly smaller and denser than those through the one-stage homogenization heat treatment. However, 410 °C was a relatively high temperature for the precipitation of $Al_3Sc$, so the precipitated grains were slightly thicker than the $Al_3Zr$ grains precipitated from alloy A(0.1Zr). Although the $Al_3Sc$ grains of alloy B(0.05Sc) through the two-stage homogenization heat treatment were slightly smaller than those through the one-stage homogenization heat treatment, both types of homogenization were high-temperature heat treatments for the $Al_3Sc$ grains. Consequently, the changing range (~10%) of the $Al_3Sc$ grain sizes of alloy B(0.05Sc) after the two types of homogenization treatments was much lower than that (~40%) of the $Al_3Zr$ grains of alloy A(0.1Zr).

The microstructures of the hot-rolled alloy A(0.1Zr) containing trace Zr and alloy B(0.05Sc) containing trace Sc that were subjected to the 420 °C annealing heat treatment for 2.5 h and the 30% cold working process are shown in Figure 4. Regardless of the one-stage or two-stage homogenization heat treatment, all showed linear net textures. From the analysis of Figure 3, the precipitated $Al_3Zr$ grains of the Zr-containing alloy A(0.1Zr) through the two-stage homogenization heat treatment were finer and denser than those through the one-stage homogenization heat treatment. The fine grains restrained dislocations better than the coarse grains [18]. Therefore, the net textures through the

two-stage homogenization heat treatment in Figure 4c were finer than those through the one-stage homogenization heat treatment in Figure 4a; the Sc-containing alloy B(0.05Sc) also had a similar phenomenon as shown in Figure 4b,d. Regardless of the one-stage or two-stage homogenization heat treatment, the net textures of the B(0.05Sc) alloys in Figure 4b,d were denser than those of the A(0.1Zr) alloy in Figure 4a,c. Although the $Al_3Sc$ grains were slightly coarser and contained less content than the $Al_3Zr$ grains, the $Al_3Sc$ grains could restrain the dislocation movement more obviously [12].

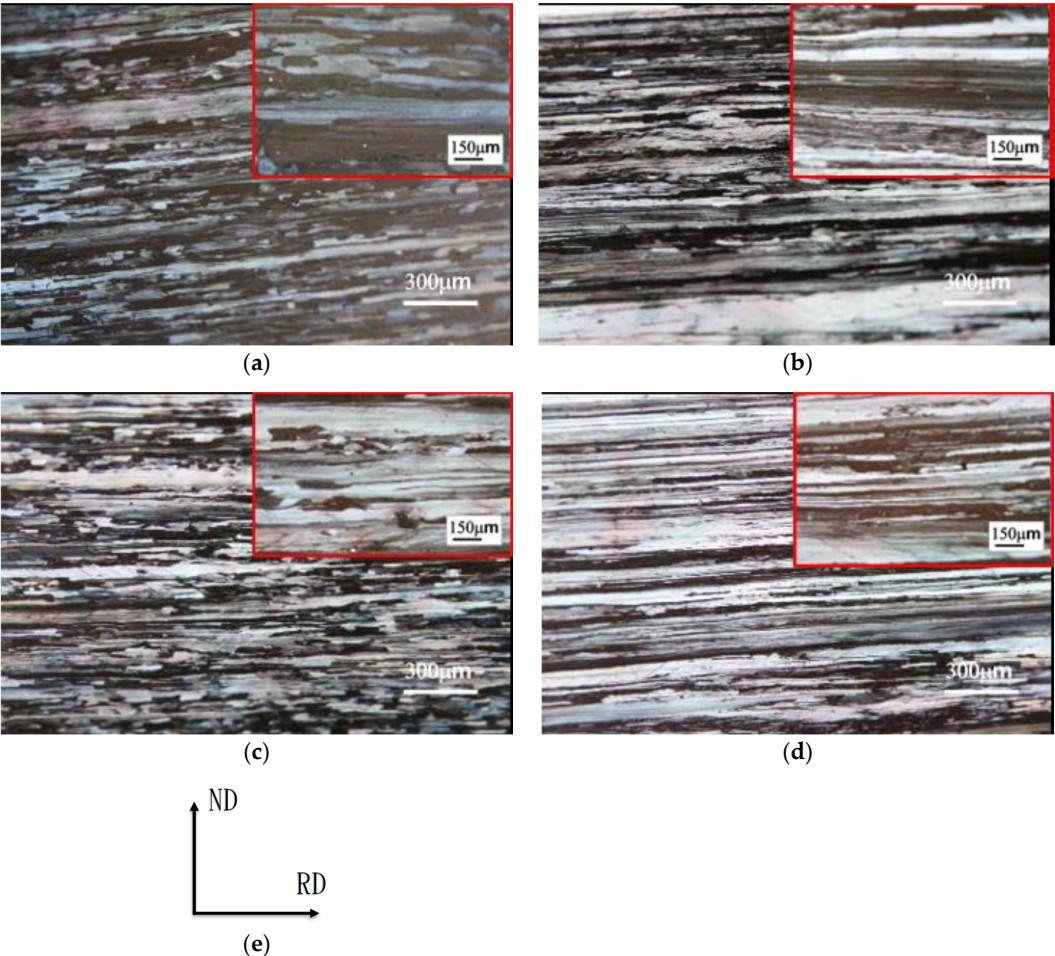

**Figure 4.** Cross-sectional microstructures of as-cold-rolled specimens observed by optical microscope (OM); (**a**) 1-Hom. alloy A(0.1Zr); (**b**) 1-Hom. alloy B(0.05Sc); (**c**) 2-Hom. alloy A(0.1Zr); (**d**) 2-Hom. alloy B(0.05Sc); (**e**) cross-sectional direction. ND, normal direction; RD, rolling direction.

From the microstructures of alloy A(0.1Zr) after T6 treatment (470 °C solution water quenching for 1 h and 120 °C artificial aging for 24 h) as shown in Figure 5a,c, a considerable amount of recrystallization and remaining textures along the rolling direction (RD) were observed. The microstructures of alloy B(0.05Sc) was also measured as shown in Figure 5b,d. Electron backscatter diffraction software (EBSD, Oxford AZtec) was used to perform quantitative analysis of the grain boundary angle distribution ratio and to calculate the degree of the recrystallization of the T6 alloys. It was found that the degree of the recrystallization of alloy B(0.05Sc) was far lower than that of alloy A(0.1Zr). After the one-stage and the two-stage homogenization heat treatments, the recrystallization ratios were about 66% and 59%, respectively. As shown in Table 2 and Figure 6a,c, alloy A(0.1Zr) after the two-stage homogenization could restrain the recrystallization of the T6 alloy more effectively than after the one-stage homogenization. This result can also be attributed to the phenomenon caused by the significantly refined $Al_3Zr$ dispersed grains of the two-stage homogenized

alloy. However, after different homogenization heat treatments, because the difference of the dispersed $Al_3Sc$ grain size of alloy B(0.05Sc) was smaller (~10%), the degrees of the recrystallization of their T6 alloys had no obvious difference. The recrystallization ratios were both about 33%, as shown in Table 2 and Figure 6b,d.

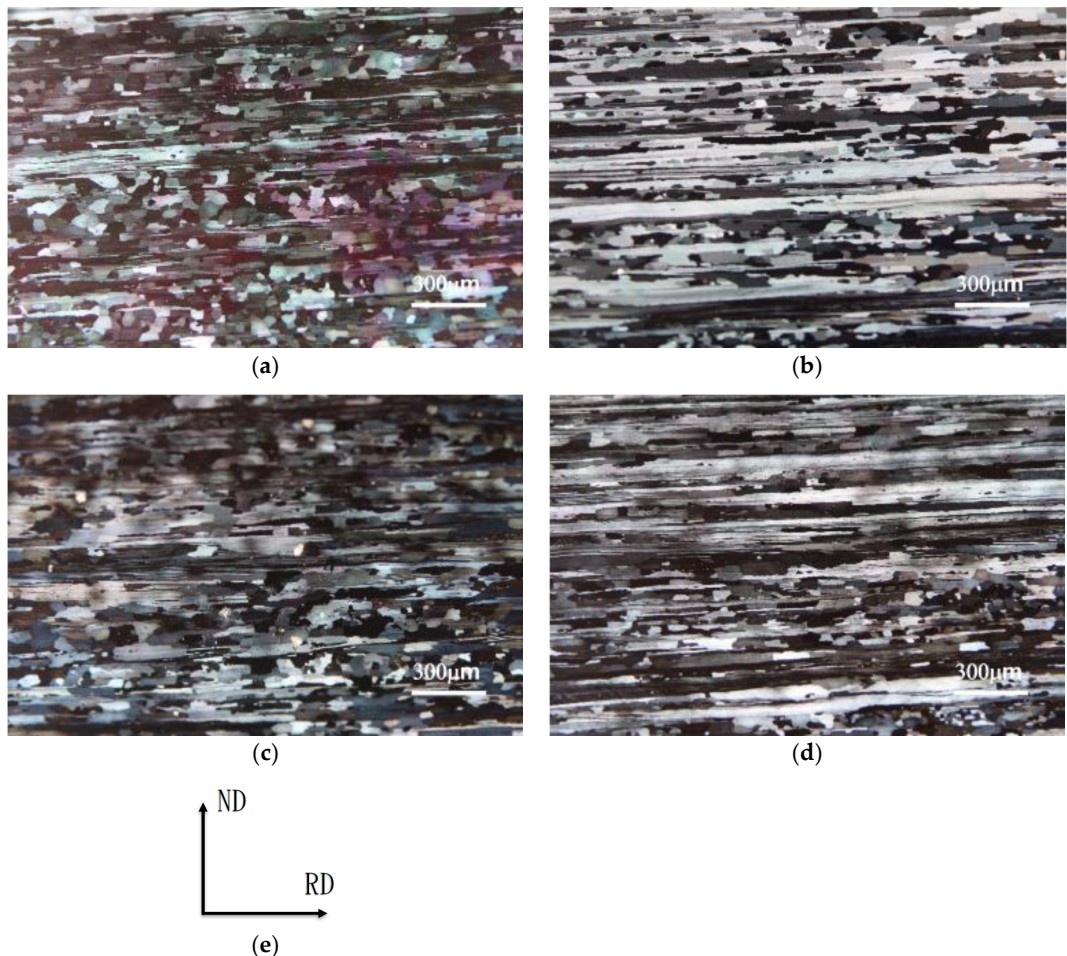

**Figure 5.** Microstructures observation by OM after T6 temper (1 h solution treatment and aging): (**a**) 1-Hom. alloy A(0.1Zr); (**b**) 1-Hom. alloy B(0.05Sc); (**c**) 2-Hom. alloy A(0.1Zr); (**d**) 2-Hom. alloy B(0.05Sc); (**e**) cross-sectional direction. ND, normal direction; RD, rolling direction.

**Table 2.** Grain size, aspect ratio and recrystallization fraction by different homogenization after T6 heat treatment of alloy A and alloy B.

| Grain Measurement Alloys and Treatment | | RD (μm) | ND (μm) | Aspect Ratio | Recrystallization Fraction (%) |
|---|---|---|---|---|---|
| Alloy A(0.1Zr) | 1-Hom.(a) | 78.3 (20.5) | 17.4 (6.0) | 4.5 (1.5) | 66.0% |
| | 2-Hom.(b) | 70.0 (36.3) | 15.9 (4.5) | 4.4 (0.9) | 59.0% |
| | [(b − a)/a] ∗ 100% | −11.00% (0.05) | −8.60% (0.15) | | |
| Alloy B(0.05Sc) | 1-Hom.(a) | 40.7 (10.2) | 14.6 (1.5) | 2.8 (0.6) | 34.0% |
| | 2-Hom.(b) | 38.8 (20.3) | 14.3 (2.1) | 2.7 (0.5) | 32.0% |
| | [(b − a)/a] ∗ 100% | −4.60% (0.22) | −2.00% (0.16) | | |

(), standard deviation; RD, rolling direction; ND, normal direction; aspect ratio: ratio of length of rolling direction and normal direction.

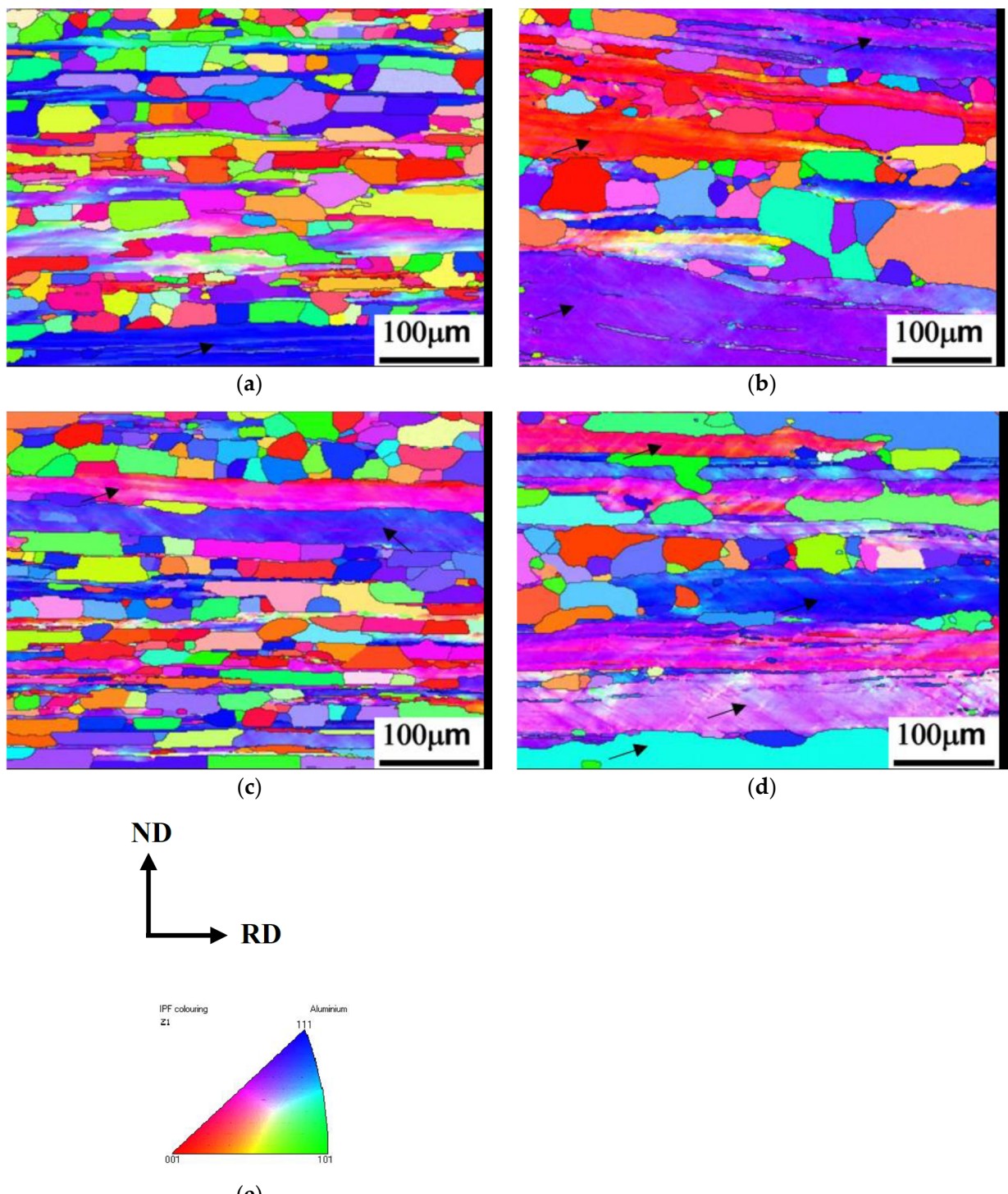

**Figure 6.** Euler angle colored electron backscattered diffraction (EBSD) maps of T6 state of (**a**) 1-Hom. alloy A(0.1Zr); (**b**) 1-Hom. alloy B(0.05Sc); (**c**) 2-Hom. alloy A(0.1Zr); (**d**) 2-Hom. alloy B(0.05Sc); (**e**) IPF(Inverse Pole Figure) coloring map and cross-sectional direction. ND, normal direction; RD, rolling direction.

The analysis of the grains of the T6 alloys in Figure 6 shows that the grains had a clear preferred orientation. The grains tended to grow in the rolling direction (RD). This phenomenon was because after the cold working, the fine high-temperature stable Al$_3$Zr and Al$_3$Sc dispersed grains arranged along the texture, and restrained the grains from growing toward the alignment direction (ND) of the texture [23,24]. After a careful analysis, the directional crystals were composed of recrystallized directional crystals and

coarse woven crystals. The coarse grains in Figure 6 (as shown by the arrow in the figure) analyzed by the Channel5 (Tango) software were actually the un-recrystallized woven crystal microstructures left after the cold working process. The grains contained many dense sub-grain boundaries. The intercept method was used to calculate the grain diameters of the recrystallized directional grains. In the summary in Table 2, it can be seen that the T6 recrystallized grain aspect ratios of the A(0.1Zr) alloy and the B(0.05Sc) alloy, whether after the one-stage or two-stage homogenization heat treatments, their T6 recrystallized grain aspect ratios are approximately the same, about 4.4 and 2.7, respectively.

From Table 2 and Figure 6a,c, it can be seen that grains of alloy A(0.1Zr) after the two-stage homogenization heat treatment were significantly finer than those after the one-stage homogenization heat treatment (RD and ND were about 11% and 8.6% smaller, respectively), showing that the $Al_3Zr$ dispersed grain phase through the two-stage homogenization heat treatment had a stronger ability to block grain growth than the grain phase through the one-stage homogenization heat treatment. For alloy B(0.05Sc), after the two-stage homogenization heat treatment, the recrystallized grains in the T6 state in Figure 6d were also thinner than those in Figure 6b after one-stage homogenization heat treatment (RD and ND were about 4.6% and 2.0% smaller, respectively), but the difference was not as obvious as that of alloy A(0.1Zr). This is because that the grain diameter of the $Al_3Sc$ dispersed phase after the two-stage homogenization heat treatment (Figure 3d) was only slightly smaller than that after the one-step homogenization heat treatment (Figure 3b).

In addition, Figure 7a,c show that when the solid solution time of alloy A(0.1Zr) increased to 24 h, the T6 alloys had almost completely recrystallized (more than 90%). It was also confirmed in Figure 7a,c, when the solution time was 1 h, that a large part of the directional grains were left over by the cold-worked textures. Not all of the directional grains were caused by the fine dispersed $Al_3Zr$ grains after the recrystallization. By comparison, as shown in Figure 7b,d, alloy B(0.05Sc), through a solution treatment for 24 h, had more recrystallization grains than through a solution treatment for 1 h. Again, it proves that part of the textures in Figure 7b,d were the leftover after the cold working. For another comparison, when the solution time was greatly increased, regardless of the one-stage or two-stage homogenization heat treatment of alloy A(0.1Zr), the previous texture was almost invisible, while the texture of alloy B(0.05Sc) was still faintly visible. This demonstrates again that in the alloy solid solution, scandium restrained the recrystallization more significantly than zirconium.

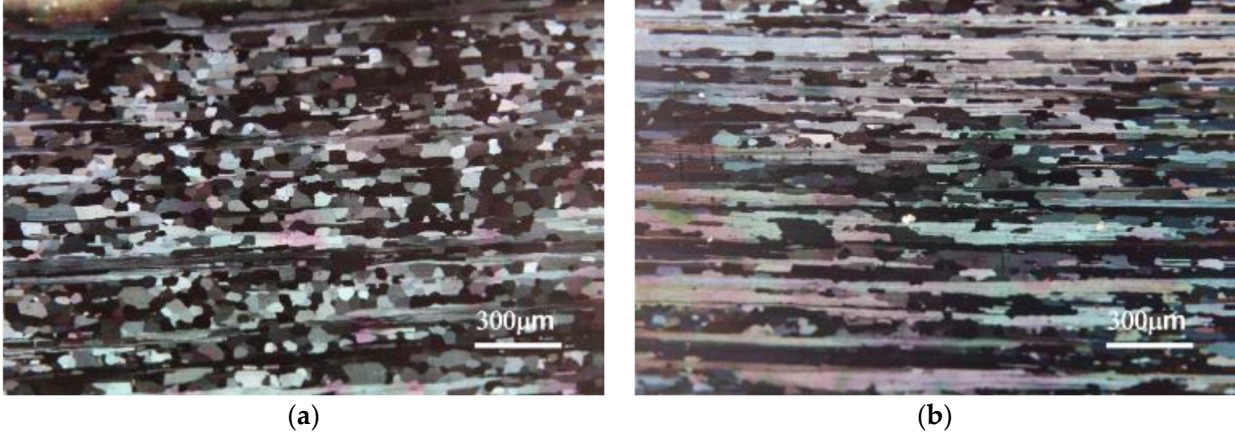

(**a**)            (**b**)

**Figure 7.** *Cont.*

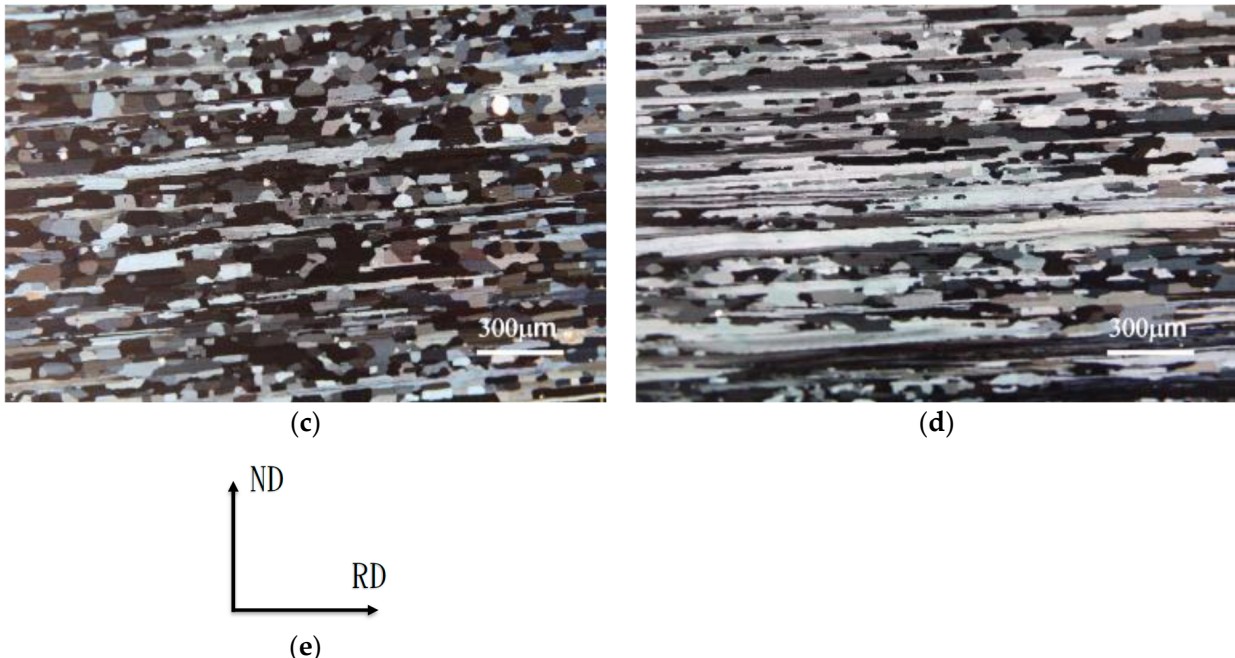

**Figure 7.** Microstructures observation by OM after 24 h solution treatment and aging: (**a**) 1-Hom. alloy A(0.1Zr); (**b**) 1-Hom. alloy B(0.05Sc); (**c**) 2-Hom. alloy A(0.1Zr); (**d**) 2-Hom. alloy B(0.05Sc); (**e**) cross-sectional direction. ND, normal direction; RD, rolling direction.

*3.2. DSC Analysis*

Figure 8 shows the DSC measurement curves of alloy A(0.1Zr) and alloy B(0.05Sc) through different homogenization heat treatments in the solid solution quenching state. The DSC test results were used to explore the precipitations of alloy A(0.1Zr) and alloy B(0.05Sc). The first exothermic peak in the figure was the precipitation of the main strengthening phase GP-zone/η′ phase, and the second exothermic peak was the transition of the η′ phase to the η phase [25,26]. In Figure 8a, it can be observed that the peaks of the GP-zone and the metastable η′ phases of alloy A(0.1Zr) after the two-stage homogenization heat treatment were more significant than those after the one-stage homogenization heat treatment. That is because different homogenization heat treatments affected the microstructures in the grains. The two-stage homogenization reduced the recrystallization ratio, increased the amount of sub-grain boundaries, and obtained a larger hardness value after the aging process [19,27].

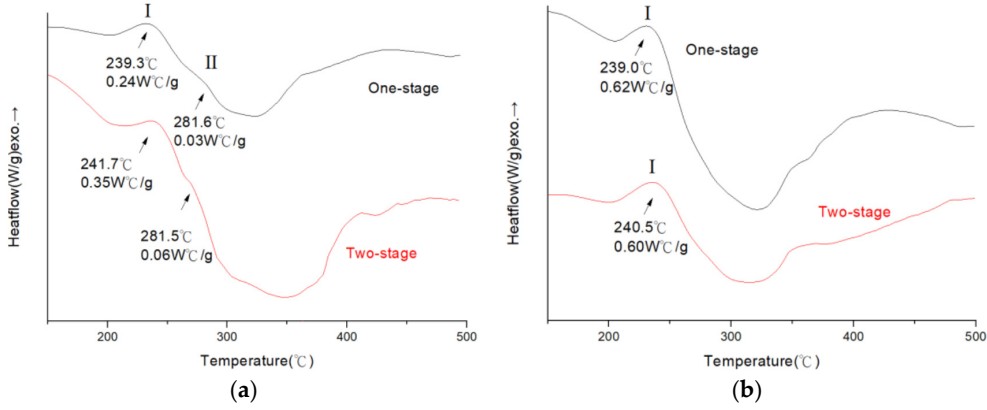

**Figure 8.** The DSC curve analysis after solution treatment of (**a**) alloy A(0.1Zr); (**b**) alloy B(0.05Sc).

After the two-stage homogenization, the peaks of the GP-zone and metastable η′ phases of alloy B(0.05Sc) were not as obvious as those of alloy A(0.1Zr), as shown in Figure 8b. This was probably related to the smaller range of sizes of the $Al_3Sc$ grains. It can also be found that, regardless of the one-stage homogenization or the two-stage homogenization, the peaks of GP-zone and metastable η′ phases of alloy B(0.05Sc) were also higher than those of alloy A(0.1Zr), showing that the Sc element reduced the precipitation rate of the strengthening phase η′ phase more effectively than the Zr element. Therefore, if the precipitation amount of the metastable state η′ phase was more, the precipitation amount of the stable state η phase became less. These findings conformed to the test results of the aforementioned microstructures and the subsequent mechanical properties.

### 3.3. Mechanical Properties

From the discussion of the microstructures, it can be learned that the Al–4.5Zn–1.5Mg alloys which were subjected to the two-stage homogenization heat treatment could obtain finer $Al_3Zr$ and $Al_3Sc$ dispersed grain phases than those that underwent the two-stage homogenization heat treatment. The Al–4.5Zn–1.5Mg alloys also could restrain the recrystallization ratios of the alloys in the T6 state. The remaining textures made the mechanical strength of the alloy higher than the recrystallized grains, but it reduced the ductility of the alloy. From the DSC analysis, it can be inferred that the Sc-containing alloy had more precipitation amount of the metastable state η′ phase, so its precipitation strengthening was better, and the mechanical strength of the alloy was improved.

Consequently, the mechanical strength of the alloys through the two-stage homogenization heat treatment was higher than that through the one-stage homogenization heat treatment. The Al–4.5Zn–1.5Mg alloys were doped with different elements and subjected to different heat treatments. Their mechanical properties are summarized in Table 3. Regardless of alloy A(0.1Zr) or alloy B(0.05Sc), their T6 hardness, yield strength and tensile strength after the two-stage homogenization heat treatment were better than those after the one-stage homogenization heat treatment, with about 2.7~4% and 1.5~2.7% increases, respectively. However, the ductility decreased by 5.7% and 1.1%, respectively.

**Table 3.** Al–Zn–Mg mechanical properties of alloy A(0.1Zr) and alloy B(0.05Sc) by different homogenization conditions.

| Heat Treatment Alloy | Mechanical Properties | One-Stage Homogenization *(a)* | Two-Stage Homogenization *(b)* | (b − a)/a ∗ 100% |
|---|---|---|---|---|
| Alloy A(0.1Zr) | Hardness (HV) | 126.7 (1.2) | 131.8 (1.7) | 4.00% (0.10) |
|  | UTS (MPa) * | 226.7 (1.7) | 232.8 (1.1) | 2.70% (0.05) |
|  | YS (MPa) | 364.5 (2.7) | 375.6 (1.9) | 3.10% (0.07) |
|  | EL (%) | 10.5 (0.5) | 9.9 (0.7) | −5.70% (0.20) |
| Alloy B(0.05Sc) | Hardness (HV) | 143.3 (1.3) | 147.1 (2.1) | 2.70% (0.06) |
|  | UTS (MPa) * | 238.3 (1.7) | 242.0 (1.9) | 1.50% (0.10) |
|  | YS (MPa) | 384.3 (2.8) | 390.5 (3.2) | 1.60% (0.02) |
|  | EL (%) | 8.9 (0.6) | 8.8 (0.3) | −1.10% (0.04) |

* ( ) Standard deviation, UTS: ultimate tensile strength, YS: yield strength, EL%: elongation.

In addition, after different homogenization heat treatments, the recrystallization degree (~34%) of alloy B(0.05Sc) in the T6 state was significantly lower than that of alloy A(0.1Zr). Therefore, from Table 3, it can also be clearly seen that the strength and hardness of alloy B(0.05Sc) were higher than those of alloy A(0.1Zr), while the ductility of alloy B(0.05Sc) was lower. It shows that the mechanical strength of alloy B(0.05Sc) with a small amount of Sc (0.05 wt%) was better enhanced than that of alloy A(0.1Zr) with a relatively high content of Zr(0.1 wt%).

The tensile fracture surface of the T6 alloy is shown in Figure 9. Whether it was alloy A(0.1Zr) or alloy B(0.05Sc), through the one-stage or two-stage homogenization heat treatment, most of the fracture surfaces of the alloys were tough dimple-like fractures. A few brittle fracture cleavages were also observed. With the careful observation and analysis

of the sizes of the dimple-like structures as shown in Figure 9d, alloy B(0.05Sc) had the smallest dimple-like structure and its cleavages were more obvious through the two-stage homogenization heat treatment. By contrast, through the one-stage homogenization, alloy A(0.1Zr) had the thickest dimples, as shown in Figure 9a, and the sizes of the other two dimple-like fractures were medium, as shown in Figure 9b,c. The test results of these mechanical properties were consistent with the observation of the microstructures.

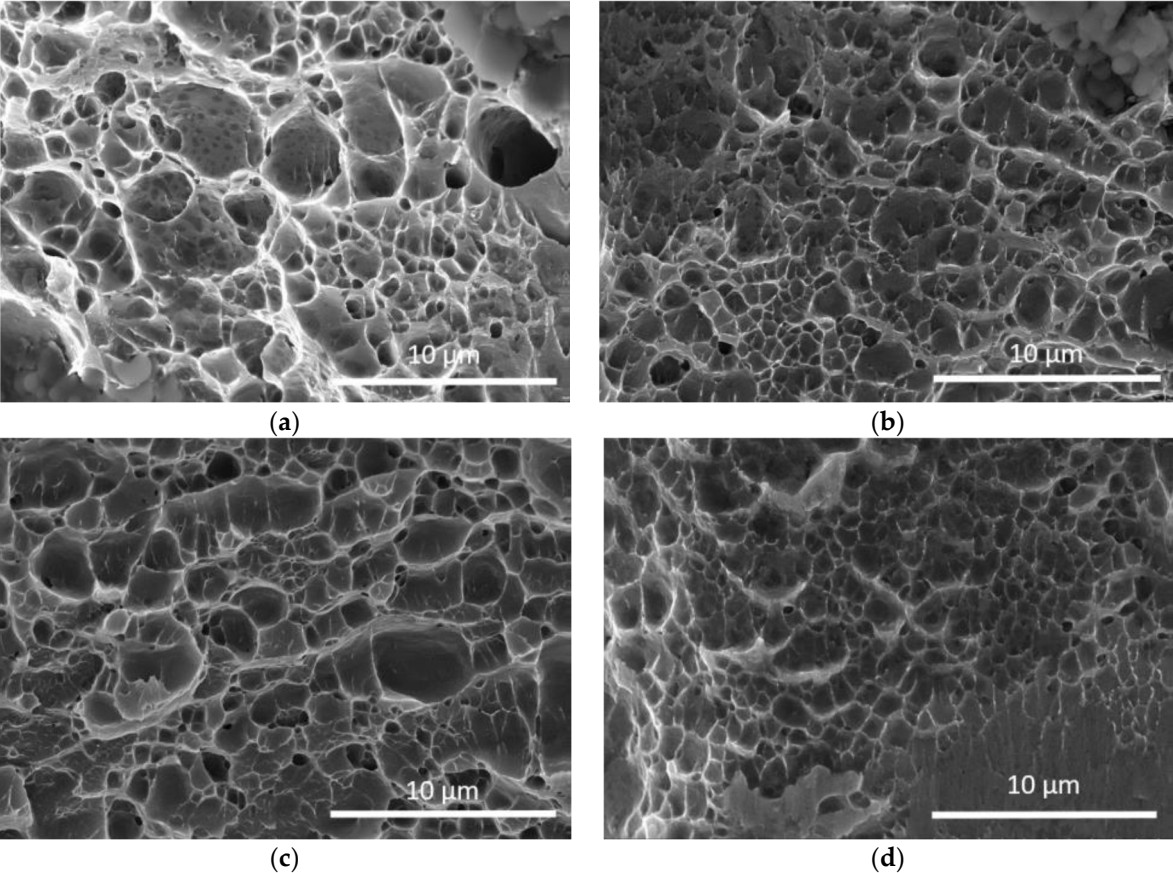

**Figure 9.** Fracture surfaces observation by SEM of T6 tempered: (**a**) 1-Hom. alloy A(0.1Zr); (**b**) 1-Hom. alloy B(0.05Sc); (**c**) 2-Hom. alloy A(0.1Zr); (**d**) 2-Hom. alloy B(0.05Sc).

## 4. Conclusions

In this study, the effects of trace transition alloy elements and two kinds of homogenization on the mechanical properties and the recrystallized microstructures of Al–4.5Zn–1.5Mg alloys were explored through the microstructure observation and the measurement of mechanical properties. The results are summarized as follows:

1.  The grain refinement effect of the as-cast, homogenized and T6 recrystallized grains of the alloy containing 0.05Sc was more obvious than that of the alloy containing 0.1Zr. Through the homogenization heat treatment, most of the as-cast dendrite structures of Al–4.5Zn–1.5Mg alloys containing Zr and Sc had been eliminated.
2.  After the homogenization heat treatment, the Al–4.5Zn–1.5Mg alloys containing Zr and Sc precipitated high-temperature thermally stable $Al_3Zr$ and $Al_3Sc$ grain phases dispersed in the aluminum matrix. The dispersed grain phases had the effect of suppressing the recrystallization and grain growth. Although the $Al_3Zr$ grains were finer than the $Al_3Sc$ grains, the dispersed $Al_3Sc$ grain phases of the Sc alloy still had a better ability to suppress T6 recrystallization and grain growth than the $Al_3Zr$ dispersed grain phase of the Zr alloy.

3. Compared with the one-stage homogenization, the two-stage homogenization made the $Al_3Zr$ and $Al_3Sc$ dispersed grains finer and denser. Therefore, the mechanical strength of the T6 alloys through the two-stage homogenization heat treatment was better than that through the one-stage homogenization heat treatment. Through the two different kinds of homogenization, the size difference between the $Al_3Zr$ dispersed grain phases of the Al–4.5Zn–1.5Mg alloys was larger than that between the Sc alloys. As shown in the collected data, through the two-stage homogenization heat treatment, the Zr alloy improved the mechanical strength more efficiently than the Sc alloys.

4. In the T6 state, the Al–4.5Zn–1.5Mg alloy containing a trace of scandium (0.05Sc) through the two-stage homogenization heat treatment had the lowest recrystallization amount and the highest tensile strength.

**Author Contributions:** Conceptualization, S.-L.L.; Data curation, T.-A.P.; Formal analysis, T.-A.P. and S.-L.L.; Funding acquisition, H.-Y.B.; Investigation, Y.-C.C. and J.-W.Z.; Methodology, J.-W.Z.; Project administration, S.-L.L.; Resources, M.-C.C.; Software, M.-C.C.; Writing–original draft, Y.-C.C. All authors have read and agreed to the published version of the manuscript.

**Funding:** The authors gratefully acknowledge the financial support received from the Ministry of Science and Technology (R.O.C) under Contract No. 109-2221-E-008-040-.

**Acknowledgments:** Thanks to Prof. C.S. Lin and Ms. Y.T. Lee of Instrumentation Center, National Taiwan University for FEG-SEM experiments.

**Conflicts of Interest:** The authors declare no conflict of interest.

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
