# Peer review of "The Effects of Multi-Stage Homogenizations on the Microstructures and Mechanical Properties of Al–Zn–Mg–Zr–Sc Alloys"

_applsci, doi:10.3390/app11020470_

Round 1

Reviewer 1 Report

This paper is very strong and fruitful. Some corrections and spellcheck in English is required, but it can be accepted as is.

Author Response

Dear Reviewer:

Thank you for your precious comments concerning our manuscript entitled “The Effects of multi-stage Homogenizations on the Microstructures, and Mechanical Properties of Al-Zn-Mg-Zr-Sc Alloys” (ID: applsci-1049563). Those comments are all valuable and very helpful for revising and improving our paper, as well as the important guiding significance to our researches. We have studied the comments carefully and have made corrections which we hope will meet with your approval. The modifications and supplements are marked in red in the revised paper. The main corrections in the paper and the responses to the comments are as follows:

Reviewer1

This paper is very strong and fruitful. Some corrections and spellcheck in English is required, but it can be accepted as is.

Response:

Thank you for your compliments and suggestions. The grammar and terminology of the manuscript has been revised by English experts. The reviewer is sincerely requested to review it again.

It is considerate of you to provide such detailed guidance. We sincerely hope that our responses will be satisfactory. Please inform us if you have any further comments or suggestions on the manuscript.

Special thanks for your constructive comments.

Reviewer 2 Report

In this study, the effects of trace transition alloy elements alloys were explored through the microstructure observation and the measurement of mechanical properties. There were studied two kinds of homogenization on the mechanical properties and the recrystallized microstructures of Al-4.5Zn-1.5Mg. It had some novelties seeking and can be accepted after minor revision.

1)The English needs to be improved.

2 ) Reorganization of the article, especially Introduction and Materials and Methods chapters, to make it easier to read.

3) A more detailed interpretation of the mechanical properties; correlation of the mechanical properties with the results obtained from the microstructural analysis and DSC.

Author Response

Response to Reviewer 2’s Comments

Dear Reviewer:

Thank you for your precious comments concerning our manuscript entitled “The Effects of multi-stage Homogenizations on the Microstructures, and Mechanical Properties of Al-Zn-Mg-Zr-Sc Alloys” (ID: applsci-1049563). Those comments are all valuable and very helpful for revising and improving our paper, as well as the important guiding significance to our researches. We have studied the comments carefully and have made corrections which we hope will meet with your approval. The modifications and supplements are marked in red in the revised paper. The main corrections in the paper and the responses to the comments are as follows:

Reviewer2

In this study, the effects of trace transition alloy elements alloys were explored through the microstructure observation and the measurement of mechanical properties. There were studied two kinds of homogenization on the mechanical properties and the recrystallized microstructures of Al-4.5Zn-1.5Mg. It had some novelties seeking and can be accepted after minor revision. 

  1. The English needs to be improved. 

Response:

    Thank you for your suggestion. The grammar and terminology of the manuscript has been revised by English experts. The reviewer is sincerely requested to review it again.

  1. Reorganization of the article, especially Introduction and Materials and Methods chapters, to make it easier to read. 

Response:

    Thank you for your suggestion. The article has been reorganized in the chapters of “Introduction”, “Materials and Methods” and “Conclusion”. 

  1. A more detailed interpretation of the mechanical properties; correlation of the mechanical properties with the results obtained from the microstructural analysis and DSC.

Response:

Thank you for your suggestion. With the data of the results, a more detailed and ostensive description of the mechanical properties is supplemented. The supplement is in Lines 326-329.

"The remaining textures made the mechanical strength of the alloy higher than the recrystallized grains, but it reduced the ductility of the alloy. From the DSC analysis, it can be known that the Sc-containing alloy had more precipitation amount of the metastable state η' phase, so its precipitation strengthening was better and the mechanical strength of the alloy was improved".

Special thanks for your constructive comments.

Reviewer 3 Report

See the attached document.

Author Response

Response to Reviewer 3’s Comments

Dear Reviewers:

    Thank you for your precious comments concerning our manuscript entitled “The Effects of multi-stage Homogenizations on the Microstructures, and Mechanical Properties of Al-Zn-Mg-Zr-Sc Alloys” (ID: applsci-1049563). Those comments are all valuable and very helpful for revising and improving our paper, as well as the important guiding significance to our researches. We have studied the comments carefully and have made corrections which we hope will meet with your approval. The modifications and supplements are marked in red in the revised paper. The main corrections in the paper and the responses to the comments are as follows:

Reviewer3

Review of “The Effects of multi-stage Homogenizations on the Microstructures, and Mechanical Properties of Al-Zn-Mg-Zr-Sc Alloys”

  1. The introduction should be broken up into 2-3 paragraphs.

Response:

Thank you for your suggestion. The introduction is broken up into 4 paragraphs.

  1. Line 143-147 – What are the standard deviations of the grain size measurements using the intercept method? Are these differences statistically relevant?

Response:

    Thank you for your suggestion. The details about the standard deviations of the grain size measurements are supplemented.

  1. Line 274-284 – It is a bit hard to claim the difference between the heat treatments based on the grain size measurements without reported standard deviations to see if these values are indeed statistically different.

Response:

    Thank you for your suggestion. The details about the standard deviations of the grain size measurements are included in Table 2.

  1. Grammatical Corrections

Line 8 - Do you need the albert77918@gmail.com here? 

We have deleted it.

Line 33 - change to "forging, heat-treated, high-strength" 

Line 38 - delete "the" at the end of the line 

Line 38 - delete “although" 

Line 39 - change "alloy, it does not ..." to "alloy, but it does not ..." 

Line 55 - use subscripts for "Al3Zr and Al35c" 

Line 55-change "are selected here to do the research" to "were selected for this study" 

Line 64 - change "functions” to “effects” 

Line 64-change "When the adding amounts of Zr and Sc are the same, certain effects of 65 the Al3SC grains are even more excellent" to "When the adding equal amounts of Zr and Sc, certain effects of the Al3Sc grains are enhanced” 

Line 66 – change “high temperature lower than the" to "high temperature, which is below the" 

Line 71 - replace “It” with "This" 

Line 82 - replace "with a view to" to "with the intention of” 

Line 91 - replace "was" with "were” 

Line 102 - Table 1 title should be left aligned 

Line 129 - change "alloys are to be continuously explored" to "alloys will need to be studied further" 

Line 138 – change "The grain" to "The average grain" 

Line 225 - Please name the software used for the backscatter diffraction analysis. This software is Oxford Aztec.

Line 246 – Include a space after (e) 

Line 254 – Include a space after (e) 

Line 297 - change "It shows again" to "This demonstrates" 

Line 303 - change the ";" to a "." 

Line 314 - change "smaller change range of the size of the Al3Sc grains” to “smaller range of sizes of the AlzSc grains” 

Line 340 - move the period to the right side of the 3 

Line 340 - please carry the standard deviations through to the percent differences in Table 3 

Response:

    Thank you for your revision. The grammar and terminology of the manuscript has been revised and checked. The reviewer is sincerely requested to review it again.

Special thanks for your constructive comments.
